# Protective Effects of Isostrictiniin Against High-Fat, High-Sugar Diet-Induced Steatosis in MASLD Mice via Regulation of the AMPK/SREBP-1c/ACC Pathway

**DOI:** 10.3390/nu16223876

**Published:** 2024-11-13

**Authors:** Qi Yan, Chenyang Li, Jinfeng Li, Yuhan Yao, Jun Zhao

**Affiliations:** 1School of Pharmacy, Xinjiang Medical University, Urumqi 830011, China; yyanqii@163.com; 2Xinjiang Key Laboratory for Uighur Medicine, Institute of Materia Medica of Xinjiang, Urumqi 830004, China; licy0609@126.com (C.L.); 17726840515@163.com (Y.Y.); 3School of Life Science and Technology, Xinjiang University, Urumqi 830046, China; lijinfeng0312@163.com

**Keywords:** MSALD, isostrictiniin, high-fat–high-sugar diets, AMPK/SREBP-1c/ACC pathway

## Abstract

Objectives: Isostrictiniin (ITN), a natural polyphenol extracted from *Nymphaea candida* (snow-white waterlily), has antioxidant and hepatoprotective activities that may be beneficial in treating metabolic dysfunction-associated steatotic liver disease (MASLD). This study aimed to investigate the protective effects of ITN on high-fat, high-sugar diet (HFSD)-induced steatosis in MASLD mice and its mechanisms. Methods: Kunming mice were randomly divided into normal control and HFSD groups. After being fed for 4 weeks, the HFSD group was randomly divided into model, atorvastatin calcium (ATC; 10 mg/kg), and ITN (25, 50, and 100 mg/kg) groups. After continued feeding for 4 weeks, the biochemical indexes in the mice were determined. Results: Compared with the model group, the liver index; FBG; HOMA-IR; serum AST, ALT, TG, TC, and LDL-C; and liver MDA, IL-6, TNF-α, and IL-1β levels in the ITN (25, 50, and 100 mg/kg) and ATC (10 mg/kg) groups were significantly decreased (*p* < 0.05), while serum HDL-C and liver SOD and GSH-Px levels were increased (*p* < 0.05). Pathological observation showed that ITN treatment mitigated the lipid liver deposition in the HFSD mice. Moreover, ITN could upregulate liver-tissue p-AMPK/AMPK protein expression in the HFSD-induced MASLD mice and downregulate SREBP-1c and ACC levels (*p* < 0.05). Conclusions: ITN can significantly improve MASLD mice, and its mechanism may be related to the regulation of the AMPK/SREBP-1c/ACC pathway.

## 1. Introduction

Metabolic dysfunction-associated steatotic liver disease (MASLD) is a steatotic liver disease (SLD) that occurs with no alcohol intake and one or more cardiometabolic risk factors. MASLD includes a range of liver diseases, such as simple hepatic steatosis, metabolic dysfunction-associated steatohepatitis (MASH), and fibrosis [1,2,3]. If not treated in time, MASH can progress to hepatic cirrhosis, even hepatocellular carcinoma (HCC). The incidence rates of MASLD represent approximately 30% of the global population and seriously harm human health. AMPK plays an important role in the regulation of lipid metabolism [4]. Activated AMPK can enhance the oxidative metabolism of fatty acids, promote lipid consumption, reduce the accumulation of fat, and improve lipid metabolism disorders by acting on its downstream targets, thus preventing and treating MASLD [5,6]. Studies have shown that lipid synthesis can be reduced by regulating the AMPK/SREBP-1c/ACC signaling pathway, thereby delaying the development of MASLD [7,8].

*Nymphaea candida* (snow-white waterlily) is the dry bud of a perennial aquatic herb and is mainly distributed in the Bohu, Yili, and Altay regions of Xinjiang, China. As a medicinal and edible plant, *N. candida* is efficacious in antipyresis and nourishment of the liver, moistening the throat and eliminating thirst, and relieving inflammation and cough, and it is used for the treatment of liver injury, hyperglycemia, hyperlipidemia, respiratory infection, and other diseases [9,10,11,12]. Polyphenols are the main characteristic components of *N. candida*, and isostrictiniin (ITN; Figure 1) has the highest content of up to 0.9%. Previous studies have shown that ITN has antioxidant, anti-inflammatory, and anti-liver fibrosis effects [13]. Based on the establishment of a mouse MASLD model, this study explored the regulatory effects of ITN on the lipid metabolism disorder of MASLD and its possible mechanism, providing an experimental basis for the development of hepatoprotective ITN-based drugs.

## 2. Materials and Methods

### 2.1. Chemicals and Reagents

Isostrictiniin was made by the phytochemical laboratory of the Xinjiang Institute of Medicine, and its purity was more than 95%. Atorvastatin calcium tablets were purchased from Pfizer Pharmaceuticals Inc. (Shanghai, China). High-fat feed was purchased from Dyets Biotechnology Co., Ltd. (Wuxi, China). ALT, AST, TG, TC, HDL-C, and LDL-C kits were purchased from Shenzhen Mindray Biomedical Technology Co., Ltd. (Shenzhen, China). IL-1β, IL-6, TNF-α, INS, SOD, MDA, and GSH-Px kits were purchased from Wuhan Elabscience Biotechnology Co., Ltd. (Wuhan, China). GA-3 Blood Glucose test strips were purchased from Sinocare (Wuhan, China). D-fructose, 5x protein loading buffer, a BCA Protein Concentration Assay Kit, RIPA buffer, and an SDS-PAGE gel preparation kit were purchased from Beijing Solarbio Science & Technology Co., Ltd. (Beijing, China). GAPDH antibody, AMPK antibody, SREBP-1c antibody, and ACC antibody were purchased from Wuhan Sanying Biotechnology Co., Ltd. Phosphorylated (*p*)-AMPK antibody was purchased from ABclonal Biotechnology (Wuhan, China). D101 microporous resin was purchased from Tianjin Xingnan Yuneng Polymer Technology Co., Ltd. (Tianjin, China). Polyamide (30–60 mesh) was purchased from the Zhejiang Taizhou Luqiaosijia biochemical plastic factory (Taizhou, China). Methanol and ethanol were commercially analytically pure.

### 2.2. Preparation of ITN

*N. candida* flowers were purchased from Xinjiang Sheng-Xiang Cao Co. in Urumqi, China and were identified by Researcher Jiang He at the Xinjiang Institute of Materia Medica, where the specimens are now stored. Following a previously reported method [13], the medicinal material (1.0 kg) was extracted with 70% ethanol by reflux to obtain a 70% ethanol extract (171.2 g). The extracts were purified by D101 macroporous resin (70% ethanol elute) and polyamide (70% ethanol elute) to obtain total polyphenols. The total polyphenols were repeatedly purified by an MCI gel CHP-20p column to obtain ITN (10.51 g). The purity of the ITN was determined to be 95.97% (Appendix A) by the HPLC normalization method according to the previous chromatographic condition [10].

### 2.3. Animals

SPF-grade male Kunming mice (18~22 g) were purchased from the Experimental Animal Center of Xinjiang Medical University. They were maintained in an SPF environment with an indoor temperature of 24~26 °C and a relative humidity of 50~60%, with light and dark alternated for 12 h.

### 2.4. Experimental Design

The Kunming mice were randomly divided into a normal control group (NC, n = 8) and a high-fat–high-sugar diet group (HFSD, n = 40). The NC group was fed with a conventional diet, and the HFSD group was fed pellets with 60% fat, 20% protein, and 20% carbohydrate (Dyets Biotechnology, Wuxi, China) and given drinking water (30% fructose) [14]. After 4 weeks, the HSFD group was randomly divided into five groups (n = 8): model (HFSD), atorvastatin calcium (ATC; 10 mg/kg), and ITN (25, 50, and 100 mg/kg) groups. The NC and HFSD groups were administered an equal volume of 0.5% CMC-Na solution via intragastric administration, and the rest were given the corresponding dose for 4 weeks. During the administration period, the NC group continued to feed on the conventional diet, and the other groups were continually fed HFSD diets. After the last administration, the mice were fasted for 12 h and weighed; then, eyeballs were removed to take blood, which then stood at room temperature for 2 h and was centrifuged at 4 °C and 3000 rpm/min for 15 min, after which the upper layers of transparent serum were taken and frozen in the refrigerator at −80 °C for examination. The liver of each mouse was taken out and weighed. The largest lobe of the liver was divided into two parts and fixed in 4% paraformaldehyde, while the remaining liver tissue was wrapped in tin foil, frozen in liquid nitrogen, and subsequently stored at −80 °C.

### 2.5. Serum Biochemical Assays

The serum AST, ALT, TG, TC, HDL-C, and LDL-C levels in mice were detected by a BS-240vet automatic biochemical analyzer (Shenzhen Mindray Animal Medical Technology Co., Ltd., Shenzhen, China).

### 2.6. Determination of FBG, FINS, and HOMA-IR

After blood collection, fasting blood glucose (FBG) was assessed using a glucose meter. Serum fasting insulin (FINS) was detected by a mouse insulin ELISA kit, and the insulin resistance index (HOMA-IR) was calculated (HOMA-IR = FBG × FINS/22.5).

### 2.7. Histological Analysis

The fixed liver-tissue samples were taken, sections of which were paraffin-embedded and treated with hematoxylin–eosin (H&E) and Oil red O staining, then sealed and examined under a light microscope.

### 2.8. Hepatic Homogenate MDA, SOD, GSH-Px, TNF-α, IL-1β, and IL-6 Levels

Hepatic tissue was homogenized with PBS at a mass-to-volume ratio of 1:9 to prepare a 10% hepatic homogenate. The MDA, SOD, TNF-α, GSH-Px, IL-6, and IL-1β levels were determined according to the instructions provided with the respective assay kits.

### 2.9. Western Blot Assay

Mouse liver tissue was taken and 900 μL high-efficiency RIPA lysate and 9 μL protease inhibitor were added for tissue homogenization at 12,000 rpm/min, followed by centrifugation at 4 °C for 15 min; then, the supernatant was obtained, protein concentration was determined by a BCA kit, protein loading buffer was added to allow denaturation at high temperature, and the samples were stored for future use. Polyacrylamide gel electrophoresis was utilized to separate the proteins, after which they were transferred to a PVDF membrane and sealed with a rapid sealing solution, followed by primary antibody addition and overnight incubation at 4 °C. The next day, the second antibody was added and incubated at room temperature for 1 h. ECL developer was then applied to exposure. Image J 1.46r software was used for strip gray-value analysis.

### 2.10. Statistical Analysis

The data were analyzed with SPSS 26.0 (IBM, Armonk, NY, USA), and the experimental results were presented as means ± standard deviations (SDs). One-way ANOVA or the Kruskal–Wallis test was utilized for data analysis, and *p* < 0.05 was considered statistically significant.

## 3. Results

### 3.1. Protective Effects of ITN on Steatosis in MASLD Mice

After 8 weeks, the HFSD group exhibited statistically significant increases in body weight and the liver index compared to the NC group (*p* < 0.01). The mice in the ITN (25, 50, and 100 mg/kg) and ATC (10 mg/kg) groups showed significant decreases in body weight and liver index compared to the HFSD group (*p* < 0.05; Figure 2 and Appendix A). Moreover, compared with the NC group, the serum TG, TC, and LDL-C levels exhibited a significant increase in the HFSD group (*p* < 0.01), while the serum HDL-C levels were statistically reduced (*p* < 0.01, Appendix A). ITN supplementation significantly improved the serum lipid-profile abnormalities induced by HFSD feeding dose-dependently (*p* < 0.05). In order to examine the potential of ITN in mitigating liver cell damage induced by HFSD, serum transaminase levels were assessed. These transaminases are well-recognized as biomarkers for evaluating liver injury. The serum ALT and AST levels in the HFSD group showed a statistically significant rise in comparison to the NC group (*p* < 0.01, Appendix A). ITN (25, 50, and 100 mg/kg) treatment reduced HFSD-induced plasma ALT and AST elevation in a dose-dependent manner (*p* < 0.05; Figure 2). The results indicated that ITN had a hepatoprotective effect.

As shown in Figure 3, the gross liver tissues in the NC mice displayed brown coloration, while the liver tissues from the HFSD group displayed noticeable white areas with ample lipid droplets, indicating significant lipid deposition in the liver (Appendix A). In contrast, there were significant improvements in the ITN (25, 50, and 100 mg/kg)-treated HFSD mice. HE staining showed that the livers of the HFSD mice displayed severe steatosis and numerous fat vacuoles compared with the NC group. ITN (25, 50, and 100 mg/kg) could significantly alleviate these pathological features: the fat vacuoles were smaller, and their numbers were decreased (Appendix A). Oil Red O staining further demonstrated that the livers of the HFSD mice contained numerous lipid droplets of varying sizes, while ITN (25, 50, and 100 mg/kg) remarkably reduced both the volume and quantity of these lipid droplets. These results indicated that ITN treatment mitigated lipid deposition in the livers of HFSD mice (Appendix A).

### 3.2. Effects of ITN on FBG, FINS, and HOMA-IR in Mice

As shown in Figure 4, the FBG, FINS, and HOMA-IR levels in the HFSD group were significantly increased compared with the NC group (*p* < 0.01). In contrast, the FBG and HOMA-IR levels in each treatment group were remarkably reduced compared to the HFSD group (*p* < 0.05); FINS was significantly decreased in the ITN (100 mg/kg) and ATC groups (10 mg/kg) (*p* < 0.05, *p* < 0.01) (Appendix A).

### 3.3. Effects of ITN on Inflammatory Cytokines and Oxidative Stress in HFSD Mice

As shown in Figure 5, compared with the NC group, the levels of IL-6, TNF-α, and IL-1β in the liver tissue of mice in the HFSD group were significantly increased (*p* < 0.01); compared with the HFSD group, the levels of IL-6, TNF-α, and IL-1β in the liver tissue of mice in each administration group were significantly decreased (*p* < 0.01, *p* < 0.05) (Appendix A). These results suggest that ITN may improve NAFLD and inhibit the expression of inflammatory factors. In addition, compared with the NC group, the SOD and GSH-Px levels in the liver tissues of mice in the HFSD group were significantly decreased (*p* < 0.01), while MDA levels were significantly increased (*p* < 0.01). Compared with the HFSD group, SOD and GSH-Px levels in the liver tissues of mice in the ITN (50 and 100 mg/kg) and ATC (10 mg/kg) groups were significantly increased (*p* < 0.05). MDA levels were significantly decreased (*p* < 0.05, Appendix A).

### 3.4. Effects of ITN on Hepatic-Tissue p-AMPK/AMPK, SREBP-1c, and ACC Protein Expression

We next sought to explore the way in which ITN attenuates HFD-induced lipid metabolism by regulating the expression levels of SREBP-1c and ACC through AMPK activation. The expressions of AMPK, p-AMPK, SREBP-1c, and ACC in liver tissues were detected. The expression of p-AMPK/AMPK protein in the liver was significantly decreased (*p* < 0.01), and the expression of SREBP-1c and ACC protein was significantly increased (Figure 6) when compared to the NC group (*p* < 0.01). The expression of p-AMPK/AMPK protein in the liver was significantly increased in each administration group (*p* < 0.05), and the protein expressions of SREBP-1c and ACC were significantly decreased when compared to the HFSD group (*p* < 0.05,) (Appendix A and Appendix A). Collectively, ITN may reduce hepatic lipid deposition by regulating the AMPK/SREBP-1c/ACC signaling pathway.

## 4. Discussion

Currently, the pathogenesis of MASLD is unclear; however, the “multiple-hit” hypothesis is widely accepted, which posits that hepatic lipid accumulation is one of the significant factors in the development of MASLD [15,16,17]. High-fat and high-sugar diets disrupt the balance between the supply, formation, and consumption of triglycerides and liver oxidation and processing. This disruption affects insulin and lipid metabolism, along with immune-related pathways; promotes the synthesis of free fatty acids; and causes lipid deposition in the liver, thus leading to insulin resistance and lipid metabolism disorders, then inflammation and oxidative stress [18,19,20,21]. Therefore, reducing lipid accumulation in the liver is an effective method for the treatment of MASLD, but there are currently no effective pharmacological treatments available, and management primarily relies on lifestyle changes, including diet and exercise. However, many MASLD patients find it challenging to sustain these lifestyle improvements. Therefore, it is crucial to enhance research on the pathogenesis of MASLD and to identify safe and effective drugs for its prevention and treatment.

Previous results showed that polyphenols from *N. candida* had better preventive and therapeutic effects on diabetic mice induced by ALX or high-fat, high-sugar diets compared with STZ-induced T2DM mice [11]. As a main characteristic ingredient, ITN may have preventive and therapeutic effects on MASLD. The aggravating effects of high-fat–high-sugar diets on sugar–lipid metabolism can induce an increase in hepatic fat deposition, hepatic triglyceride accumulation, and insulin resistance, which then lead to liver injury [22]. In this experiment, the changes in these relevant indicators in HFSD mice indicated that the MASLD model was successfully established. After the interventions with ITN (50 and 100 mg/kg) and ATC (10 mg/kg), the blood lipids, insulin resistance, and liver function in the HFSD mice were significantly improved: liver steatosis and lipid deposition were significantly reduced in the MASLD mice by ITN.

Hepatic lipid overload induced by lipid metabolism disorders can induce overproduction of reactive oxygen species (ROS). Overexpression of ROS can cause oxidative modification of cellular macromolecules (DNA, lipids, proteins, etc.), resulting in the accumulation of damaged macromolecules and subsequent liver damage [23]. ITN possesses better antioxidative and hepatoprotective effects. In this study, ITN could significantly decrease liver-tissue MDA levels in HFSD mice as well as improve liver-tissue SOD and GSH-Px levels. Pro-inflammatory cytokines, particularly IL-1β, IL-6, and TNF-α, are known to be highly expressed in MASLD, especially in MASH [24]. The serum levels of IL-1β, IL-6, and TNF-α in the HFSD mice were significantly decreased by ITN.

AMPK plays a role as a master switch in regulating the homeostasis of energy metabolism in the body and can regulate lipid metabolism-related genes to keep the metabolism and synthesis of lipids at a relatively stable level [25,26,27,28]. Studies have shown that activated AMPK can inhibit ACC activity by downregulating the expression of SREBP-1c, thereby inhibiting cholesterol and fatty acid synthesis, reducing hepatic lipid synthesis, and improving hepatic steatosis [29,30,31]. The results showed that ITN could increase the expression of p-AMPK and decrease the expression of SREBP-1c and ACC, suggesting that ITN may reduce hepatic lipid deposition by regulating the AMPK/SREBP-1c/ACC signaling pathway, thus improving MASLD in mice.

## 5. Conclusions

In conclusion, our data suggest that ITN, the main active ingredient in the edible and medicinal plant *N. candida*, could effectively inhibit the development and hepatotoxicity of MASLD fatty liver induced by an HFSD diet and inhibit abnormal lipid accumulation by regulation of the AMPK/SREBP-1c/ACC signaling pathway. These results can provide a basis for the prevention and treatment of MASLD and the application of ITN.

## Figures and Tables

**Figure 1 nutrients-16-03876-f001:**
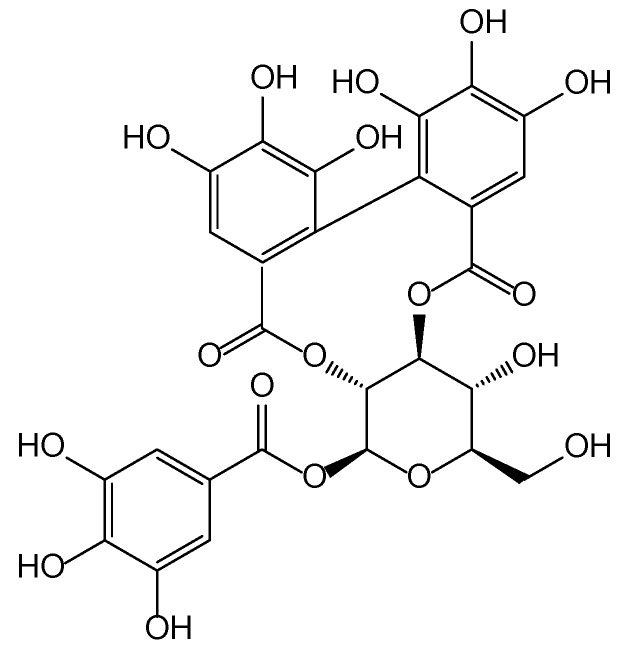
Chemical structure of isostrictiniin.

**Figure 2 nutrients-16-03876-f002:**
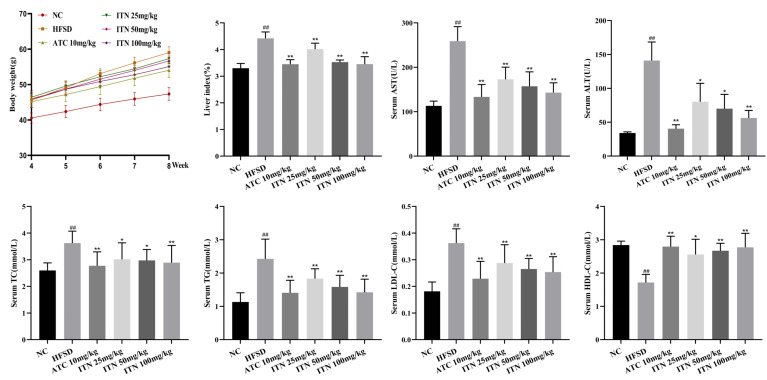
Protective effects of ITN on steatosis in MASLD mice. Data are presented as means ± SDs (n = 8). ^##^
*p* < 0.01 vs. NC; * *p* < 0.05, ** *p* < 0.01 vs. HFSD.

**Figure 3 nutrients-16-03876-f003:**
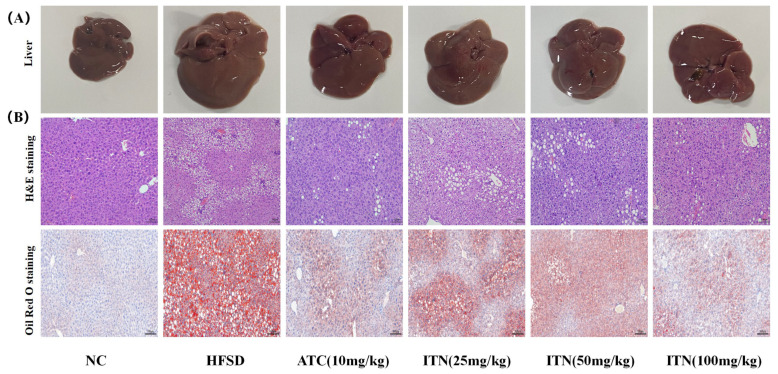
(**A**) Representative liver tissue samples of the designated groups of mice. (**B**) H&E and Oil red O staining of liver sections of mice in the indicator group. Scale, bar, 100 μm.

**Figure 4 nutrients-16-03876-f004:**
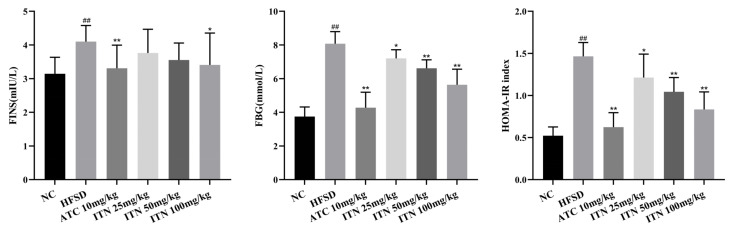
Effects of ITN on FBG, FINS, and HOMA-IR in Mice. Data are presented as means ± SDs (n = 8). ^##^
*p* < 0.01 vs. NC; * *p* < 0.05, ** *p* < 0.01 vs. HFSD.

**Figure 5 nutrients-16-03876-f005:**
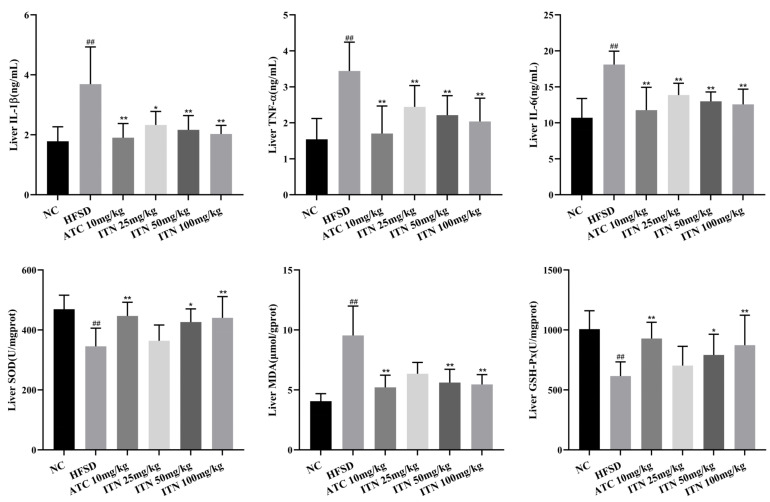
Effects of ITN on inflammatory cytokines and oxidative stress in HFSD mice. Data are presented as means ± SDs (n = 8). ^##^
*p* < 0.01 vs. NC; * *p* < 0.05, ** *p* < 0.01 vs. HFSD.

**Figure 6 nutrients-16-03876-f006:**
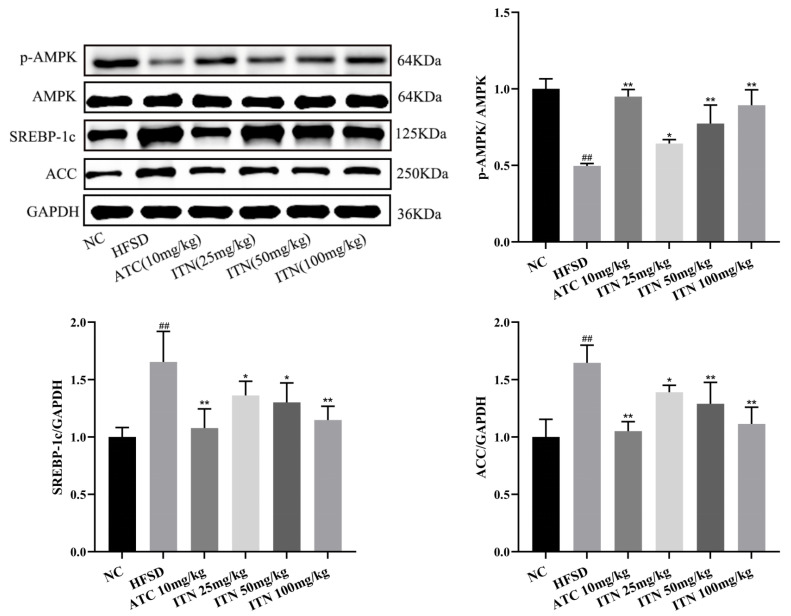
Effects of ITN on the hepatic-tissue AMPK/SREBP-1c /ACC pathway in mice. Data are presented as means ± SDs (n = 3). ^##^
*p* < 0.01 vs. NC; * *p* < 0.05, ** *p* < 0.01 vs. HFSD.

## Data Availability

Data supporting this study are included in the article and Appendix A.

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
