# Peer review of "Protective Effects of Isostrictiniin Against High-Fat, High-Sugar Diet-Induced Steatosis in MASLD Mice via Regulation of the AMPK/SREBP-1c/ACC Pathway"

_nutrients, 2024, doi:10.3390/nu16223876_

Round 1

Reviewer 1 Report

Comments and Suggestions for Authors

In this paper, the authors describe the role of isostrictiniin in NAFLD mice. This paper is interesting and scientifically sound with clearly presented results. However, there are major grammatical and language issues, along with typos that make the paper less readable. For example:

- "This study aimed to invesrigate..."

- "steatosis in on NAFLD mice" 

- "while the expression of SREBP-1c and ACC protein were significantly decreased"

- Repeated typos, such as "decreasement", "dsicussion"....

- "ITN posses the better antioxidantive and hepatotective effects..."

- Some uncommon terms like "heat" or "dry heat" are used: "N. candida posses the efficacies of reducing heat and nourishing liver, moistening throat and eliminating thirst, dampening brain and relieving inflammation and cough, and is used for the treatment of dry heat induced liver deficiency, upset and thirst and other diseases."

- A "normal group" is mentioned. Please clarify the type of group used, should it be referred to as the "control group"? 

Also, NAFLD is an older term so please try using terms like "MASLD" or "MAFLD", instead.

Comments on the Quality of English Language

There are significant errors in English language that make the paper difficult to read.

Author Response

Dear Editors and Reviewer 1:

Thank you for your comments on our manuscript entitled “Protective Effects of Isostrictiniin against High-Fat, High-Sugar Diet-Induced Steatosis in NAFLD Mice via Regulating AMPK/ SREBP-1c/ACC Pathway”. These comments are of great help to the revision and improvement of this paper, and also have the important guiding significance of our research. We have studied the ideas carefully, Now the title is changed to “Protective Effects of Isostrictiniin against High-Fat, High-Sugar Diet-Induced Steatosis in MASLD Mice via Regulating AMPK/ SREBP-1c/ACC Pathway”, and an amendment has been made to the article in the hope of approval. The revised portion is marked with yellow highlight in the manuscript. The main corrections in the manuscript and the response to the reviewer’s comments have been expressed as follows:

Comment 1: However, there are major grammatical and language issues, along with typos that make the paper less readable. For example:"This study aimed to invesrigate...""steatosis in on NAFLD mice""while the expression of SREBP-1c and ACC protein were significantly decreased"Repeated typos, such as "decreasement", "dsicussion"...."ITN posses the better antioxidantive and hepatotective effects..."

Response: Thank you very much for the reviewer's valuable suggestion. We have corrected some wrong words and grammar in the article, and the modified contents are highlighted in yellow.

Comment 2: Some uncommon terms like "heat" or "dry heat" are used: "N. candida posses the efficacies of reducing heat and nourishing liver, moistening throat and eliminating thirst, dampening brain and relieving inflammation and cough, and is used for the treatment of dry heat induced liver deficiency, upset and thirst and other diseases."

Response: Thank you very much for the reviewer's valuable suggestion. These contents have been revised for the readability of the manuscript.

Comment 3: - A "normal group" is mentioned. Please clarify the type of group used, should it be referred to as the "control group"?

Response: Thank you very much for the reviewer's valuable suggestion. This "normal group" should be called " normal control group", This has been modified in the text, and the modified contents are highlighted in yellow.

Comment 4: Also, NAFLD is an older term so please try using terms like "MASLD" or "MAFLD", instead.

Response: Thank you very much for the reviewer's valuable suggestion. By reviewing the literature, we found that MASLD was the new term and replaced NAFLD with MASLD. The modified contents are highlighted in yellow.

Special thanks to you for your good comments.

Reviewer 2 Report

Comments and Suggestions for Authors

General comments:

The manuscript is short and relatively easy to read, but it is presented in a rather sloppy manner in such matters as spacing, formatting of figures, lack of data about equipment and reagent manufacturers, etc.

In addition to this, it is somewhat strange that after the methods of production that the authors have described (very generic) they have achieved, as they cite, a degree of purity of 95.97% in a single polyphenol. I am not an expert in the plant used as a source (Nymphaea candida), but I have obtained polyphenols from plants and marine algae and normally the number of polyphenols that can be obtained from a species is usually high, so it is somewhat strange that such a high purity is obtained with such simple methods of production.

Similarity index of the manuscript (Excluding references) is 45%. From then, 23% corresponds to the article published by the same research group entitled “Preventive Effect of the Total Polyphenols from Nymphaea candida on Sepsis-Induced Acute Lung Injury in Mice via Gut Microbiota and NLRP3, TLR-4/NF-κB Pathway”, recently published in International Journal of Molecular Sciences. In both cases, the indexes are excessive and must be dramatically decreased

In all the manuscript, results showing statistically significant differences are generically mentioned as P<0.05 or P<0.01. The specific value obtained in all comparisons should always be cited and not generic values.

Specific comments:

Page 1, line 13: Please correct “invesrigate”

Page 2, line 43: The first reference about snow-white waterlity is in this line, and it results strange to the reader, some mention should have been made in the title or in the abstract.

Page 2, line 50: “Nymphaea candida” was previously abbreviated as “N. candida”. It should be cited in its abbreviated form.

Page 2, lines 63-64: Please cite the manufacturers and addresses of both microporous resin and polyamide, as well as all regents employed for the assays.

Page 2, line 66: More information is required about HPLC method used for determine  the polyphenols purity.

Page 2, line 68: Please define “SPF”.

Page 3, line 93: Biochemical analyzer…What specifically?

Lines 124-126: This is not a result, it is a repetition of those cited in materials and methods section.

Page 4-7: Spaces after the images are out of format. Please correct it.

Page 4 and all manuscripts. In the main text, it is cited “Fig.”, and in the footnotes are cited as “Figure”. Please, be consistent.

Author Response

Dear Editors and Reviewer 2:

Thank you for your comments on our manuscript entitled “Protective Effects of Isostrictiniin against High-Fat, High-Sugar Diet-Induced Steatosis in NAFLD Mice via Regulating AMPK/ SREBP-1c/ACC Pathway”. These comments are of great help to the revision and improvement of this paper, and also have the important guiding significance of our research. We have studied the ideas carefully, Now the title is changed to “Protective Effects of Isostrictiniin against High-Fat, High-Sugar Diet-Induced Steatosis in MASLD Mice via Regulating AMPK/ SREBP-1c/ACC Pathway”, and an amendment has been made to the article in the hope of approval. The revised portion is marked with yellow highlight in the manuscript. The main corrections in the manuscript and the responds to the reviewer’s comments have been expressed as follows:

Comment 1: Page 1, line 13: Please correct “invesrigate”

Response: Thank you very much for the reviewer's valuable suggestion. We modified it and the modified contents are highlighted in yellow.

Comment 2: Page 2, line 43: The first reference about snow-white waterlity is in this line, and it results strange to the reader, some mention should have been made in the title or in the abstract.

Response: Thank you very much for the reviewer's valuable suggestion. We supplemented it in the abstract section, and the supplemented content is marked in yellow.

Comment 3: Page 2, line 50: “Nymphaea candida” was previously abbreviated as “N. candida”. It should be cited in its abbreviated form.

Response: Thank you very much for the reviewer's valuable suggestion. “Nymphaea candida” was abbreviated as “N. candida”, and the modified contents are highlighted in yellow.

Comment 4: Page 2, lines 63-64: Please cite the manufacturers and addresses of both microporous resin and polyamide, as well as all regents employed for the assays.

Response: Thank you very much for the reviewer's valuable suggestion. D101 microporous resin and polyamide (30-60 mesh) were purchased from Tianjin Xingnan Yuneng Polymer Technology Co (China) and Zhejiang Taizhou Luqiaosijia biochemical plastic factory (China) respectively; methanol and ethanol were commercially analytically pure. These are supplemented by chemical and reagents in the text,and the modified contents are highlighted in yellow.

Comment 5: Page 2, line 66: More information is required about HPLC method used for determine the polyphenols purity.

Response: Thank you very much for the reviewer's valuable suggestion. The HPLC conditions were as follows: Phenomenex Gemini-NX C18 column (250 mm×4.6 mm, 5μm), the mobile phase of A(acetonitrile) and B(0.2%phosphoric acid, v/v) with a gradient elution (0~35min, 5%-15%A; 35~65 min, 15%-18%A; 65~70min, 18%-20%A; 70~75 min, 20%-5%A), flow rate at 1.0 mL/min, column temperature at 30℃, and detection wave length at 266 nm. The purity of ITN was 95.94%, 95.97% and 95.99% by HPLC normalization method. This will be provided in the supporting materials.

Comment 6: Page 2, line 68: Please define “SPF”.

Response: Thank you very much for the reviewer's valuable suggestion. The SPF stands for Specific pathogen Free. SPF grade animal refers to the experimental animals that do not carry the main potential infection or conditional pathogenic bacteria or pathogens that interfere with scientific experiments except the pathogens that should be excluded from the cleaning animals.

Comment 7: Page 3, line 93: Biochemical analyzer…What specifically?

Response: Thank you very much for the reviewer's valuable suggestion. Automatic biochemical analyzer was a measuring instrument for biochemical index, and BS-240vet automatic biochemical instrument mentioned in manuscript was purchased from Shenzhen Mindray Animal Medical Technology Co., Ltd in China. This is supplemented in the text, and the modified contents are highlighted in yellow.

Comment 8: Lines 124-126: This is not a result, it is a repetition of those cited in materials and methods section.

Response: Thank you very much for the reviewer's valuable suggestion. We revised this part of the manuscript.

Comment 9: Page 4-7: Spaces after the images are out of format. Please correct it.

Response: Thank you for your careful review and for pointing out the formatting issues in our manuscript. We appreciate your attention to detail and agree that adherence to the journal’s formatting guidelines is crucial for the clarity and presentation of our work. We have adjusted the format of the article.

Comment 10: Page 4 and all manuscripts. In the main text, it is cited “Fig.”, and in the footnotes are cited as “Figure”. Please, be consistent.

Response: Thank you very much for the reviewer's valuable suggestion. This part has been revised in the original manuscript, and the modifications are marked in yellow.

Special thanks to you for your good comments.

Round 2

Reviewer 2 Report

Comments and Suggestions for Authors

Most of the comments adressed by the Reviewer in the firt round or review was satifactorely required, including the decrease in the similarity index with respect to the previous article in the firt round of reviews.

However, one of the comments stated by the Reviewer in the firt round was not adressed and should be corrected previously to its aceptance. In concrete, this comment:

"In all the manuscript, results showing statistically significant differences are generically mentioned as P<0.05 or P<0.01. The specific value obtained in all comparisons should always be cited and not generic values."

Author Response

Dear Editors and Reviewer 2:

Thank you for your comments on the manuscript entitled “Protective Effects of Isostrictiniin against High-Fat, High-Sugar-Diet-induced Steatosis in NAFLD mice via Regulatory AMPK/ SREBP-1c/ACC Pathway”. These comments have been of great help in revising and improving this article and have also provided important guidance for our study. We have carefully studied these ideas and now changed the title to “Protective Effects of Isostrictiniin against High-Fat, High-Sugar-Diet-induced Steatosis in NAFLD Mice via Regulatory AMPK/ SREBP-1c/ACC Pathway” and revised the article for approval. The revised parts are highlighted in yellow in the manuscript. The main corrections in the manuscript and the responses to the reviewer's comments are expressed as follows:

Comment: "In all the manuscript, results showing statistically significant differences are generically mentioned as P<0.05 or P<0.01. The specific value obtained in all comparisons should always be cited and not generic values."

Response: Thank you very much for the reviewer's valuable suggestions. This part has been revised in the original manuscript, and the revisions are marked in yellow. 

Special thanks for your kind comments.